# OmniTry: Virtual Try-On Anything without Masks

**Yutong Feng**[1,†]     **Linlin Zhang**[2]     **Hengyuan Cao**[2]     **Yiming Chen**[1]

**Xiaoduan Feng**[1]     **Jian Cao**[1]     **Yuxiong Wu**[1]     **Bin Wang**[1,‡]

[1]Kunbyte AI   [2]Zhejiang University

{fengyutong.fyt, binwang393}@gmail.com {zhanglinlinlin, caohy}@zju.edu.cn
{chenyiming, fengxiaoduan, caojian, wuyuxiong}@k-fashionshop.com

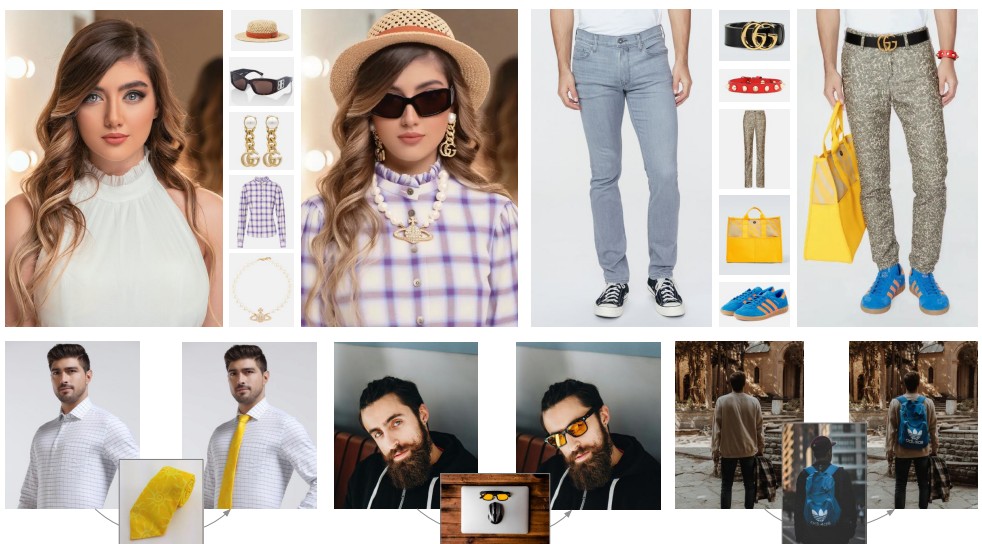

Figure 1: Try-on results of various wearable objects generated by `OmniTry`, which supports object images with white or natural backgrounds, and even try-on results as input.

## Abstract

Virtual Try-ON (VTON) is a practical and widely-applied task, for which most of existing works focus on clothes. This paper presents `OmniTry`, a unified framework that extends VTON beyond garment to encompass any wearable objects, *e.g.*, jewelries and accessories, with mask-free setting for more practical applications. When extending to various types of objects, data curation is challenging for obtaining paired images, *i.e.,* the object image and the corresponding try-on result. To tackle this problem, we propose a two-staged pipeline: For the first stage, we leverage large-scale unpaired images, *i.e.,* portraits with any wearable items, to train the model for mask-free localization. Specifically, we repurpose the inpainting model to automatically draw objects in suitable positions given an empty mask. For the second stage, the model is further fine-tuned with paired images to transfer the consistency of object appearance. We observed

---

[†]Project Leader. [‡]Corresponding Author.

that the model after the first stage shows quick convergence even with few paired samples. `OmniTry` is evaluated on a comprehensive benchmark consisting of 12 common classes of wearable objects, with both in-shop and in-the-wild images. Experimental results suggest that `OmniTry` shows better performance on both object localization and ID-preservation compared with existing methods. The code, model weights, and evaluation benchmark of `OmniTry` are available at https://omnitry.github.io/.

# 1 Introduction

The image-based virtual try-on (VTON) [19] has received tremendous attention due to its wide application in e-commerce. Given a person image and a garment image, the purpose of VTON is to transfer the garment onto the person as a preview. Thanks to the success of large-scale image generative models [50, 47, 14, 32] with their photorealistic aesthetics, recent efforts [60, 28, 9, 10, 66] have achieved satisfying performance on both generation quality and garment identity preservation.

Despite the advancement of VTON, existing methods mainly concentrate on clothing try-on. Though some works have explored the extension to non-clothing, such as shoes [11] and ornaments [42], there still lacks a unified framework in the literature, supporting any types of wearable objects. Furthermore, most methods require the indication of wearing area on person (*e.g.,* masks or bounding boxes), or use automatic human-body parsers [62] to identify the area. When extending to anything try-on, it would be impractical to expect users to draw the targeting area, as the interaction between the model and various objects can be more considerably more complex. It is also challenging to leverage existing parsers to detect appropriate try-on areas for diverse objects. Thus, we follow the mask-free setting [24, 16, 66] for the model to automatically localize the area with natural composition.

When confronting anything try-on, one key challenge is the data collection. Generally, the training of VTON requires large-scale *paired* images, consisting of a single-shot of the garment, and a corresponding person try-on result. Most datasets are curated from e-commerce websites, with at least thousands of samples, *e.g.,* VITON-HD [8] and DressCode [43]. While for many common types of wearable objects, such as hats and ties, there is no abundant quantity of paired data, but only the product pictures. This limitation makes it necessary to develop an efficient training framework.

In this paper, we present `OmniTry`, targeting mask-free virtual try-on for any wearable object. `OmniTry` reduces the heavy reliance on paired training samples, leveraging large-scale *unpaired* images for prior learning. The unpaired images refer to the image containing a person with any wearable objects, which can be easily obtained from existing database. The training of `OmniTry` can be separated into two stages: (i) The first stage is completely conducted on unpaired data. We use multi-modal large language models (MLLMs) [1] to list all wearable items with descriptions. Each item is detected and erased from the image, forming a training pair. Then an image generative model is trained to re-paint the item, prompted by the corresponding text description. After stage one, the model is expected to know how to transfer various objects onto the person in proper position, size and orientation. (ii) For the second stage, we further leverage high-quality paired data to fine-tune the model. Object image is introduced into the context, modulating the model to preserve the consistency of object appearance. Building upon the model from stage one, we observe that ID-consistency is quickly adapted even fine-tuned with few samples. To summarize, the two stages in `OmniTry` contributes the ability of mask-free localization and ID-preservation, respectively.

Regarding the model design, we leverage the diffusion transformer as backbone, and compare two variants, *i.e.,* text-to-image and inpainting model. Experimental results show that the inpainting model can be rapidly repurposed as a mask-free generative model, by simply setting the mask input with all-zero values. Image tokens from different images are concatenated in the sequence dimension, and processed with full-attention mechanism for consistency learning [53, 67, 7, 23]. We employ efficient adapter tuning techniques for transferring the model to this task. More specifically, we implement two distinct adapters that handle the tokens from person and object images, individually.

The erasure of wearable object is observed with critical impact. A naive solution is to call object-removal models [52, 71, 27] to fill the area of objects. However, we notice that while the processed area appears visually normal, it contains imperceptible artifacts. Thus, the model learn undesirable shortcuts by identifying these traces, resulting in poor generalization to natural images. To tackle this problem, we propose *traceless erasing* to eliminate the artifacts. We conduct image-to-image [41] to

subtly re-paint the entire image after erasure. Subsequently, the original try-on image is blended with the re-painted image, ensuring the non-object area remains unchanged. Traceless erasing disrupts the erasure boundaries, thereby compelling the model to learn genuine try-on capability.

We construct a comprehensive evaluation benchmark covering 12 common types of wearable objects, divided into clothes, shoes, jewelries and accessories. To fully investigate the model robustness, the objects are set on white and natural backgrounds, or try-on images, referring to Fig. 1. Metrics are designed to evaluate the object consistency, person preservation and wearing position. Experimental results indicate that `OmniTry` outperforms existing methods, and achieves efficient few-shot training.

## 2 Related Works

**Controllable Image Generation.** The breakthrough of diffusion model [20] has driven extensive research on controllable image generation. ControlNet [64] and related pioneering works [45, 48, 68] explore precise control with diverse conditions. IP-Adapter [63] and related studies [15, 22, 31, 34, 39] investigate online concept control to achieve subject customized generation. Recent developments in DiT [46] have further propelled generalized image generation and editing. In-context LoRA [23] enables diverse thematic generation with image concatenation. OminiControl [53] introduces task-agnostic condition control with minimal model modification. OmniGen [58] unifies multi-task processing via large vision-language models. UniReal [7] achieves unified image editing via full-attention and video data prior. VisualCloze [35] enhances visual in-context learning for cross-domain generalization. For localized image customization, Anydoor [6] pioneers to transfer subject into specified region. MimicBrush [5] extends to local components transferring with imitative editing. ACE++ [40] establishes a unified paradigm for generation and editing tasks.

**Image-based Virtual Try-On (VTON)** has emerged as a critical task attracting tremendous efforts. VITON [19] introduces Thin Plate Spline transformations [2] for multi-stage garment processing. CP-VTON [55] formalizes explicit geometric warping and texture synthesis stages. GP-VTON [59] combines local flow estimation with global parsing to improve detail preservation. These warping-based approaches, however, face persistent challenges in cross-sample alignment and generalization. This motivates the adoption of diffusion models [20], including TryOnDiffusion's parallel U-Net [70], LADIVTON's garment tokenization [44], and DCI-VTON's hybrid warping-diffusion framework [17]. OOTDiffusion [60] and FitDiT [25] enhance detail fidelity through specialized attention mechanisms. Though with advanced results, most of them remain constrained by intensive preprocessing requirements (*e.g.,* wearing masks and pose estimation). Boow-VTON [66] creates a mask-free approach through in-the-wild data augmentation. Any2AnyTryon [18] pioneers fully mask-free implementations, eliminating dependency on masks or poses.

## 3 Method

### 3.1 Preliminary

**Diffusion Transformer (DiT).** `OmniTry` is developed on DiT [46], a scalable transformer architecture for diffusion-based generation. The image is encoded into latent space through an autoencoder [29], and patchified into tokens [13]. Diffusion process [20] is conducted on tokens with a transformer consuming the noisy tokens and predicts for denoising. Recent advancement in DiT, *i.e.,* rectified flow matching [37] and rotary position embedding (RoPE) [51], are also involved in this paper.

**Virtual Try-On (VTON).** Given a person image $\mathcal{I}_P$ and a wearable object image $\mathcal{I}_O$, the try-on result image is noted as $\mathcal{I}_T$. Suppose the segmentation mask of the object in $\mathcal{I}_T$ is $\mathcal{M}$, then the target of VTON is three-fold: (i) the consistency between objects in original and try-on images, *i.e.* $\min \text{similarity}(\mathcal{I}_T \mathcal{M}, \mathcal{I}_O)$, (ii) the preservation of non-wearing areas, *i.e.,* $\mathcal{I}_T(1-\mathcal{M}) = \mathcal{I}_P(1-\mathcal{M})$, (iii) the object is properly located on person, evaluated through the quality of $\mathcal{I}_T$.

### 3.2 Stage-1: Mask-Free Localization

As illustrated in Fig. 2, the training of `OmniTry` consists of two stages, corresponding to the abilities of localization and ID-preservation, respectively. In the first stage, the objective of training can be regarded as "garment-free VTON", in contrast to the "model-free VTON" in the literature [18]. Given

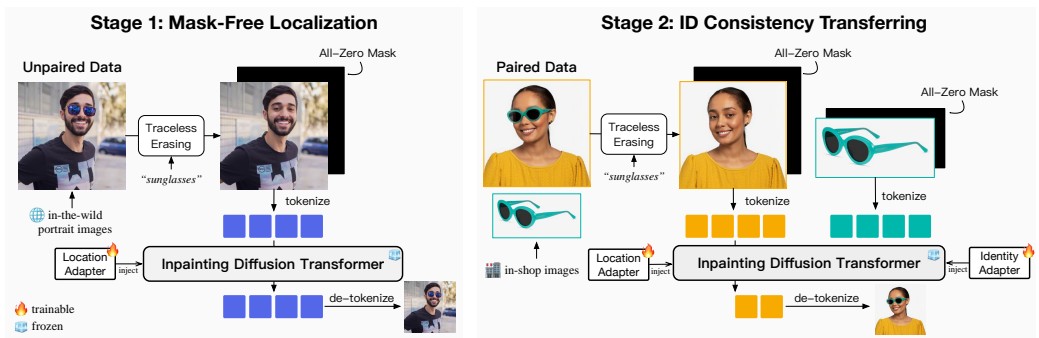

Figure 2: **The two-staged training pipeline of `OmniTry`.** The first stage is built on in-the-wild portrait images to add wearable object onto the person in mask-free manner. The second stage introduces in-shop paired images, and targets to control the consistency of object appearance.

the person image $\mathcal{I}_P$ and the object description, the model aims to edit $\mathcal{I}_P$ by adding the object as described. The type and detailed appearance of object are only prompted by input text. Control signal indicating the wearing area, *e.g.,* bounding boxes, masks or selecting point, is not introduced here. Such an objective enforces the model to concentrate on *where* to paint the object, and *how* to blend it harmoniously with the person image. The training of stage one can be easily supervised by a portrait image database, for which we introduce how to construct the training samples in the next paragraph.

**Unpaired Data Pre-process.** We refer single portrait image as *unpaired* image with only try-on result $\mathcal{I}_T$, in contrast to the paired images $(\mathcal{I}_T, \mathcal{I}_O)$ in next stage. We start by curating a large-scale dataset containing any human-related images. The dataset is filtered by a classifier for images with a person wearing at least one object. Following that, we leverage a MLLM, Qwen-VL 2.5 [1], to list all potential wearable objects in each image. The output includes both the type of object and its appearance description. We also prompt MLLM to add an interaction description, *e.g.,* "wearing sunglasses" and "holding sunglasses in hand", to distinguish various cases. To erase each object for training, we use GroundingDINO [36] and SAM [30] to obtain the object mask, and remove the object with an inpainting-based erasing model. Specifically, we fine-tune an internal erasing model based on Flux.1 Fill [32]. Though without erasing capability, it is observed to quickly adapt to this task with a few training samples. To summarize, the pre-precessing pipeline outputs a set of triples, including the original image as $\mathcal{I}_T$, the object-erased image as $\mathcal{I}_P$, and the object textual description.

**Model Architecture: Text-to-Image *v.s.* Inpainting Model.** There are two candidate variants of model to implement the mask-free try-on task, *i.e.,* the text-to-image (T2I) model, and the mask-based inpainting model. Generally, mask-based VTON models [28, 10] leverage the fill-in capacity of inpainting model, while mask-free methods [66, 18] adapt the T2I model, by injecting subject features into the backbone. Following the recent success in controllable image generation [53, 23, 67], a straightforward solution with T2I model is to concatenate the person image tokens into the sequence of noisy tokens, then processed with the full-attention mechanism in DiT. This strategy effectively transfers the person appearance into the target image, while also doubles the computation cost.

In contrast, `OmniTry` explores to repurpose the inpainting model for mask-free generation. The inpainting model is generally finetuned from the T2I model via extending the input channels. Suppose the noisy latent as $X$, the input image as $I_c$, and the inpainting mask as $M$. Then the extended input is $\mathrm{concat}(X; I_c(1 - M); M)$, where $\mathrm{concat}(\cdot)$ denotes channel-wise concatenation. For repurposing the model, we simply set $M = \mathbf{0}$, thus the input turns to be $\mathrm{concat}(X; I_c; \mathbf{0})$. At the initialization, the zero mask leads the output image directly repeating the input. Therefore, compared with T2I-based solution, the model effortlessly learns to copy the person condition, thus attentively focusing on locating the modification area. We inject a location adapter (implemented as LoRA [21]) for finetuning. In practice, the model converges rapidly to adapt the mask-free generative manner.

**Traceless Erasing.** Early experimental results suggest that the model learn unexpected shortcut. We visualize the training monitoring result in Fig. 3 (a), where we evaluate the model on erased training samples. It is shown that the output image almost perfectly recovers the position and shape of the object in ground-truth image, which indicates information leakage. We attribute the problem to the erasing model that leaves invisible traces in the filling area [56, 69]. The model tends to figure out these abnormal area for editing, instead of predicting the reasonable position. When applying to real-world images, the model frequently fails to locate the try-on area, and directly repeat the input.

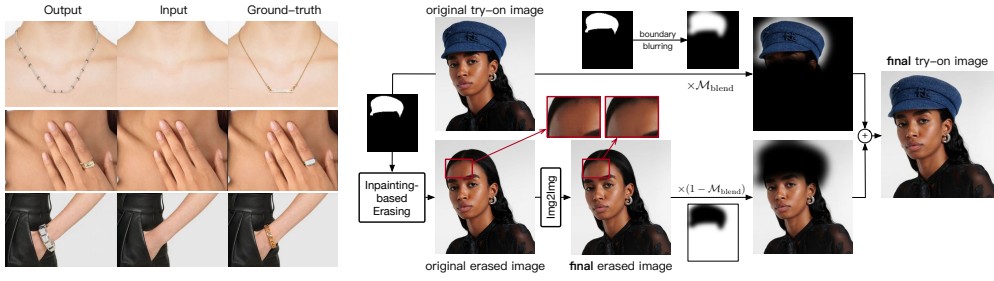

Figure 3: **Study on traceless erasing.** (a) Shortcuts learned by model with naive erasing, where the model recovers the same shape and position as the ground-truth. (b) The pipeline of traceless erasing, where image-to-image model is introduced to disturb the traces (indicated in the red boxes).

To address this problem, we propose a traceless erasing strategy, as shown in Fig. 3 (b). After erasing the object with inpainting model, we apply with an image-to-image (I2I) translation [41] to subtly re-paint the image. We first add noise to the erased image $\hat{\mathcal{I}}_P$, referring a specific timestep $t \in [0, 1]$ in diffusion schedule ($t = 0.2$ in this paper), *i.e.,* $z = enc(\hat{\mathcal{I}}_P) \times (1-t) + \epsilon \times t$, where $enc(\cdot)$ denotes the VAE encoder and $\epsilon$ is a standard Gaussian noise. Then a T2I model denoises $z$ into normal image with partial diffusion process from $t$ to $0$. In this manner, the artificial effects in inpainting area are confused with the whole re-painted image, thus avoid information leakage. Since the I2I process modifies the detail of person image, the original try-on image should be correspondingly adjusted. To achieve smooth transition in the object boundary, we modulate the original mask $\mathcal{M}$ into a blending mask $\mathcal{M}_{blend}$ by blurring the boundary area for gradual blending effect. The final try-on image is:

$$\mathcal{I}_T^{\text{blend}} = \mathcal{I}_T \times \mathcal{M}_{\text{blend}} + \text{img2img}(\hat{\mathcal{I}}_P) \times (1 - \mathcal{M}_{\text{blend}}) \tag{1}$$

### 3.3 Stage-2: ID Consistency Preservation

The second stage of `OmniTry` inherits the location adapter from stage one, and steps further to control the consistency of object appearance. Referring to Fig. 2, in-shop image pairs are leveraged containing try-on image $\mathcal{I}_T$ and object image $\mathcal{I}_O$. We pre-process the data with traceless erasing, and gather a list of triple $(\mathcal{I}_T, \mathcal{I}_P, \mathcal{I}_O)$ for training. Considering the lack of enough samples, the objective is to conduct efficient training with minimal adjustment to the model architecture in stage one.

**Masked Full-Attention.** Following the recent full-attention customization researches [53, 23], we directly append the object image tokens into the existing sequence in DiT, and shift their position embedding in the width dimension. Under this settings, OminiControl-2 [54] and EasyControl [67] also explore to block some information flow in attention. In detail, the attention mask is set to zero where the condition tokens serve as query and the generated tokens as key. Such an attention mask improves the inference efficiency, but leads to performance decrease to a certain extent.

The main difference between the above works and `OmniTry` is that the condition image is also concatenated with noisy latents and all-zero mask, for adaption to inpainting model. To cope with such variance, we design two strategies in training: (i) We compute diffusion loss on object image with itself as supervision, *i.e.,* directly copying the input, which is aligned with the zero-mask input. (ii) We block all the data flow from the generating try-on image to object image, thus avoid the detailed object appearance to be interrupted. In practice, we find it helpful to better preserve object identity with the above masked full-attention.

**Two-Stream Adapters.** To fully preserve the ability of mask-free localization, we maintain the forward process of person image tokens exactly consistent with the first stage. Then an identity adapter is initialized for the newly introduced object image tokens. The two adapters, in same architecture, serve for a two-stream computation process, *i.e.,* we switch different adapters by identifying tokens from different image sources. The inference is similar to the multi-modality DiT [14] coping with vision and language information separately.

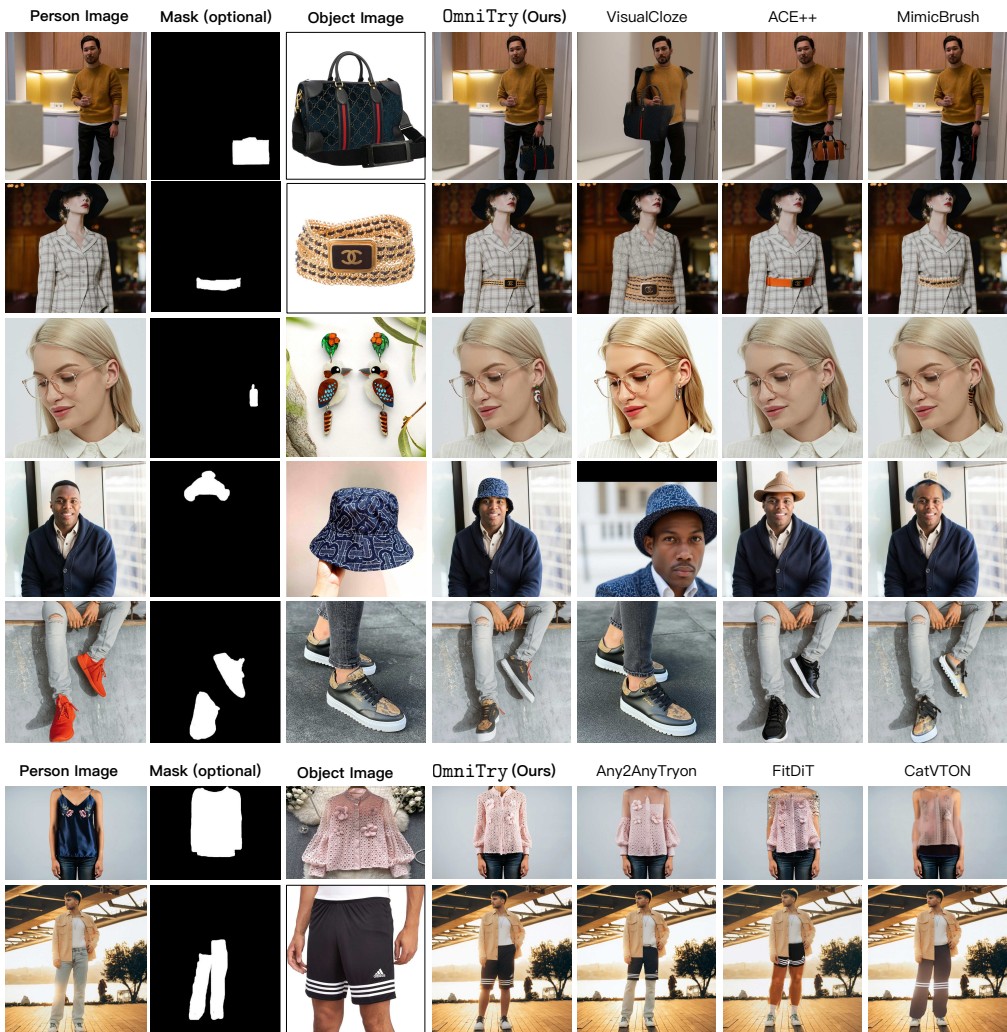

Figure 4: **Qualitative comparison** among `OmniTry` and existing methods on multiple objects.

## 3.4 Evaluation Benchmark

As the first work exploring unified virtual try-on task, we establish a comprehensive benchmark, dubbed *OmniTry-Bench*, for better evaluation and comparison with existing works.

**Benchmark Collection.** We gather evaluation samples within 12 common types of wearable objects, which can be summarized into 4 major classes: (i) *clothes* consisting of top, bottom and full-body garments, (ii) *shoes* in common styles, (iii) *jewelries*, including bracelets, earrings, necklaces and rings, (iv) *accessories*, including bags, belts, hats, glasses, sunglasses and ties. We consider detailed sub-types if necessary, such as the backpack, shoulder and tote bags. For each sub-type, we collect 15 paired test images for man and woman, separately. The object images are assigned in white background, natural background, and try-on setting, with 5 pairs for each. The person images are also set in white and natural backgrounds. Such settings ensure to fully evaluate the robustness of model. Overall, the evaluation benchmark contains 360 pairs of images.

**Evaluation Metrics.** As discussed in Sec. 3.1, the objectives of try-on can be divided into three aspects. Since there is no ground-truth result in mask-free setting, we redesign the metrics as follows:

*Object Consistency*: We crop the objects from the try-on and object images via masking, and compute the visual similarity using DINO [3] and CLIP [49], with metrics noted as M-DINO and M-CLIP-I.

*Person Preservation*: In contrast, we crop out the person from try-on and person images, and compute spatial-aligned similarity between them, *i.e.,* LPIPS [65] and SSIM [57].

Table 1: **Evaluation results on OmniTry-Bench**, which is separated into two groups: results on the whole set and the clothes subset, for fair comparison with methods only optimized on clothes data.

| method | mask | Object Consistency | | Person Presevation | | Object Localization | |
|---|---|---|---|---|---|---|---|
| | | M-DINO ↑ | M-CLIP-I ↑ | LPIPS ↓ | SSIM↑ | G-Acc. ↑ | CLIP-T ↑ |
| *on the whole set* | | | | | | | |
| Paint-by-Example [61] | | 0.4565 | 0.7727 | 0.3903 | 0.8033 | 0.9861 | 0.2804 |
| MimicBrush [5] | ✓ | 0.4693 | 0.7253 | 0.3033 | 0.8575 | 0.9250 | 0.2781 |
| ACE++ [40] | | 0.4565 | 0.7474 | 0.4561 | 0.7519 | 0.9667 | 0.2791 |
| OneDiffusion [33] | | 0.4731 | 0.7749 | 0.7001 | 0.5831 | **0.9972** | 0.2309 |
| VisualCloze [35] | ✗ | 0.5292 | 0.7782 | 0.4471 | 0.6190 | 0.9639 | 0.2524 |
| OmniGen [58] | | 0.5435 | 0.7869 | 0.6703 | 0.5965 | 0.9944 | 0.2535 |
| OmniTry (**Ours**) | | **0.6160** | **0.8327** | **0.0542** | **0.9333** | **0.9972** | **0.2831** |
| *on the clothes subset* | | | | | | | |
| Magic Clothing [4] | | 0.5665 | 0.7634 | 0.2761 | 0.8786 | 1.0 | 0.2700 |
| CatVTON [10] | ✓ | 0.5744 | 0.7906 | 0.2084 | 0.8828 | 1.0 | 0.2797 |
| OOTDiffusion [60] | | 0.5961 | 0.8016 | 0.2178 | 0.8865 | 1.0 | 0.2761 |
| FitDiT [25] | | 0.6733 | 0.8340 | 0.1618 | 0.9027 | 1.0 | 0.2831 |
| Any2AnyTryon [18] | ✗ | 0.6747 | 0.8537 | 0.2089 | 0.8969 | 1.0 | **0.2832** |
| OmniTry (**Ours**) | | **0.6995** | **0.8560** | **0.1021** | **0.9105** | **1.0** | 0.2799 |

*Object Localization*: (i) Counting the success rate whether a visual grounding model [36] detects the object, denoted as G-Accuracy. (ii) Computing the image-text similarity, noted as CLIP-I, between try-on image and a text describing the person trying on the object (generated by MLLM [1]).

## 4 Experiment

### 4.1 Experimental Setup

**Training Data.** For the first stage, we gather a diverse dataset containing both in-the-wild portrait images and in-shop model shots. Considering each image could contain multiple wearable objects, the total amount of training pairs is $188,694$. For the second stage, we collect paired samples following the 12 basic types in our benchmark. The whole dataset contains $51,195$ pairs, which shows class-unbalanced distribution ($14,861$ pairs for clothes and $295$ for ties). During training, each pair is equipped with a brief text description, such as "trying on sunglasses", to help distinguishing different classes. We note that the clothes and shoes are not erased but replaced with another one. Thus, we exchange the prefix as "replacing" for their prompts.

**Implementation Details.** We train the first stage with batch-size of 32 for 50K steps, and the second stage with batch-size of 16 for 25K steps. All the experiments are conducted on 4 NVIDIA H800 GPUs. The location and identity adapters are implemented as LoRA [21] with rank 16. We employ the AdamW [38] optimizer with learning rate of $1^{-4}$ and weight decay of $0.01$. All the images are resized to a maximum of 1 million pixels while preserving their original aspect ratios to training.

**Compared Methods.** We primarily compare with methods in two basic paradigms:

*Image-based Virtual Try-On*: Most VTON methods focus exclusively on garments. We compare on the clothes subset with representative works, including CatVTON [10], OOTDiffusion [60], Magic Clothing [4], FitDiT [25], and Any2AnyTryon [18] (the only open-sourced mask-free model).

*General Customized Image Generation*: Recent works explore to unify customization-related tasks into a single model, *e.g.,* transferring the whole subject or local components, in mask-based or mask-free manners. We compare with notable implementations, including Paint-by-Example [61], MimicBrush [5], ACE++ [40], OneDiffusion [33], OmniGen [58] and VisualCloze [35].

To cope with the methods requiring masks of editing areas, we manually draw the masks in person image regarding the type of objects. Thus, the results of these methods are listed for reference, instead of direct comparison with the remaining mask-free methods.

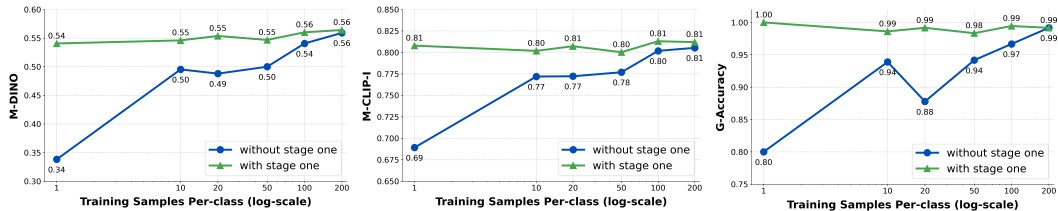

Figure 5: Ablation study on the two-staged training framework in **few-shot settings**. We show the evaluation metrics given varying amounts of paired training samples.

## 4.2    Results on Unified Virtual Try-on

**Qualitative Results.** We visualize the try-on examples generate by representative compared methods in Fig. 4. For the general customization methods, mask-based works only modify the given areas, but show unstable object identity transferring. While for the mask-free works, the results tend to be a free combination of the input person and object. Though with better consistency, they fail to precisely preserve the person image. `OmniTry` achieves accurate object consistency, in the meanwhile only edits the proper try-on areas of person image in mask-free manner. On the clothes subset evaluation, we observe that the existing VTON methods show unnatural output when evaluated on in-the-wild data. `OmniTry` is empowered by the compounded training on both in-the-wild and in-shop data, and shows more generalized ability of adapting various styles of garments.

**Quantitative Results.** Tab. 1 incorporates the evaluation results on the proposed OmniTry-Bench, conducted on the whole benchmark and the clothes subset, respectively. `OmniTry` outperforms existing methods on both sets. For the mask-based customization methods, though the input mask helps to localize the editing area, they sometimes fail to transfer the complete appearance of objects, resulting in lower consistency metrics. For the generalized customization methods in mask-free manner, they achieve better subject-ID preservation, but suffers to maintain the person image, thus show worse LPIPS and SSIM. Such quantitative results are consistent with the visualized comparison results. When evaluated on clothes subset, though `OmniTry` is not specifically optimized on clothes dataset, it still shows advancing performance compared with state-of-the-art works in mask-based and mask-free settings. We note that the mask-free try-on could not be evaluated on previous benchmarks (*e.g.,* VITON-HD [8]) for the missing of person images.

## 4.3    Ablation Study

**On the Training Strategy.** We study one of the key designs in `OmniTry`, *i.e.,* the two-staged training framework. The first stage is intended to leverage large-scale unpaired data, and boost the training efficiency in the second stage. To demonstrate this, we evaluate models initialized by the first stage and from scratch, respectively. For the comparison of efficiency, the models are fine-tuned in few-shot settings, ranging from 1 to 200 training samples per class. The results with representative metrics are illustrated in Fig. 5. For metrics related to person preservation (LPIPS and SSIM), we note that they could be higher when the model fails and directly repeats the input, thus not included.

It is observed that model from scratch shows increasing performance with more training samples per class. While for model initialized from the first stage, it already achieves satisfying performance even with only one example for training. The results demonstrate that the first stage training significantly boosts the efficiency for fine-tuning, and is especially friendly to uncommon types of objects. It is noted that though the few-shot tuning achieves good performance, we still fine-tune it with all available paired data to further increase the stability of model, referring to the results in Tab. 1

Table 2: Ablation study on the model architecture and erasing strategies of `OmniTry`.

| method | M-DINO ↑ | M-CLIP-I ↑ | LPIPS ↓ | CLIP-T ↑ |
|---|---|---|---|---|
| *on the model architecutre (the whole subset)* | | | | |
| Full Method | **0.5991** | **0.8272** | 0.0557 | 0.2830 |
| - txt2img model | 0.5005 | 0.7727 | 0.0676 | 0.2767 |
| - w.o. object loss | 0.5851 | 0.8222 | 0.0420 | 0.2824 |
| - full attention | 0.5752 | 0.8130 | **0.0384** | **0.2832** |
| - one-stream adapter | 0.5840 | 0.8186 | 0.0502 | 0.2802 |
| *on the erasing strategy (the jewelry subset)* | | | | |
| naive erasing | 0.4964 | 0.7554 | 0.0413 | 0.2727 |
| traceless erasing | **0.5389** | **0.7782** | **0.0288** | **0.2732** |

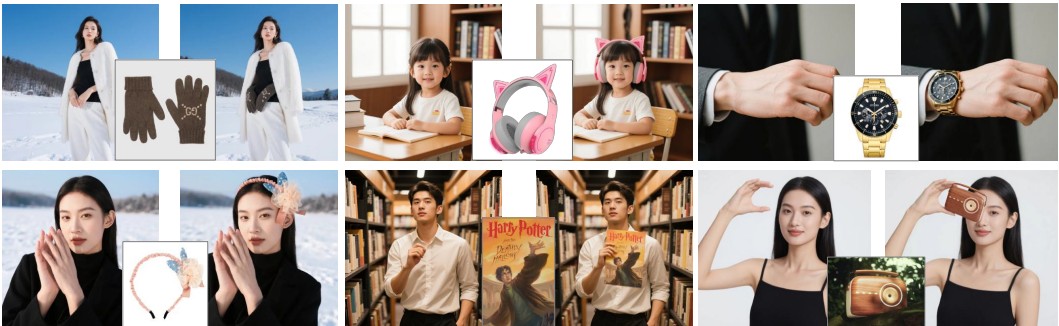

Figure 6: Try-on results of `OmniTry` fine-tuned on uncommon classes of wearable or holdable objects.

**On the Model Architecture.** We then conduct ablation study on all the explored design of model architecture in `OmniTry`. The results are shown in Tab. 2, where the "full method" indicates the final solution. (i) We start with the comparison using text-to-image and inpainting model as backbone. Results show that the inpainting backbone performs better on all metrics which is consistent with our assumption that inpainting model takes no efforts to preserve the original image and converges faster. (ii) For the additional loss computation on object image, we observe that removing the loss decreases the model performance to a certain extent. (iii) For the attention mechanism, full attention additionally introduces flow from person to object image, thus the object consistency metrics decrease correspondingly. (iv) We also investigate to use a single adapter for this task, *i.e.* applying the adapter from the first stage to all image tokens. The one-stream framework also decreases the model performance, since it plays different roles in the inference of person and object images.

**On the Traceless Erasing.** To verify the effectiveness of traceless erasing, we conduct ablation study on the jewelry subset with naive and traceless erasing. Results in Tab. 2 suggest that removing the traceless erasing leads to dramatic decrease in all metrics. Therefore, we adopt traceless erasing as a fundamental pre-processing strategy in `OmniTry`.

### 4.4 Extension to Uncommon Classes

We evaluate `OmniTry` on 12 common types of objects in the main experiments. To further demonstrate the efficiency of `OmniTry`, we extend it to some uncommon types, for which the paired training samples are limited to be obtained. The experiment is conducted on types including gloves, earphones, watches, hairbands, books and electronic products, with roughly 20 samples per class. It is noted that some types like books are actually in broader definition of try-on, *i.e.,* holdable items.

The visualization results are shown in Fig. 6. Thanks to the generalized training of the first stage, though with few paired samples, `OmniTry` succeeds in transferring these relatively uncommon objects onto the correct position. The results encourage broader extension of `OmniTry` into more application scenarios, without preparing a large amount of paired images.

## 5 Limitations

In this section, we discuss the limitations of `OmniTry` observed in practice. As the first work exploring unified VTON, `OmniTry` is still restricted by the object types in training dataset. For the efficient tuning in stage-2, it could be challenging to extend to uncommon objects not involved in the unpaired dataset in stage-1. Larger pre-training dataset is expected to further boost the generalization ability. For the mainly-focused 12 common types, experimental results show that `OmniTry` could also fail to transfer the object consistency or output poor appearance in some cases, especially for the objects with larger transformation, *e.g.,* bags. The above limitations encourage future works to build upon `OmniTry` and develop more advanced models towards unified try-on task.

# 6   Conclusion

This paper presents `OmniTry`, a unified mask-free framework extending the existing garment try-on into any wearable objects. To tackle the problem of lacking abundant paired samples, *i.e.,* object and the try-on image, for many types of objects, we propose a two-staged training pipeline in `OmniTry`. During the first stage, large-scale unpaired images are leveraged to supervise the model for mask-free object localization. While the second stage tames the model to maintain the object consistency. We elaborate the design of `OmniTry`, including a traceless erasing for avoiding shortcut learning, an inpainting-based re-purposing strategy for mask-free generation, and a masked full-attention for identity transferring. A new benchmark targeting unified try-on is introduced, and demonstrates the effectiveness of `OmniTry` compared with existing methods. Extensive experiments also verify that `OmniTry` achieves efficient learning even with few paired images for training.

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

# A  Details of Benchmark and Metrics

As the pioneering work investigating the unified virtual try-on task, we construct a comprehensive evaluation benchmark named *OmniTry-Bench*, accompanied by six dedicated metrics to systematically assess the quality of synthesized try-on images.

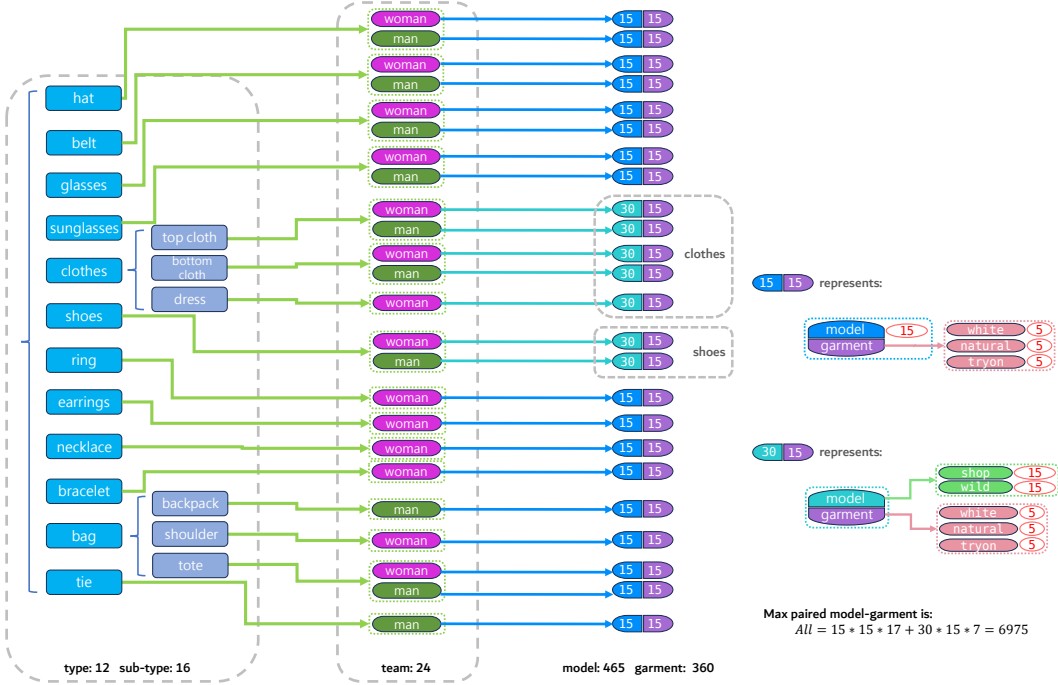

Figure 7: The visualization of the *OmniTry-Bench* constitution.

## A.1  Constitution of Benchmark

As the Figure 7, we gather evaluation samples within 12 common types of wearable objects, which can be summarized into 4 major classes: (i) *clothes* consisting of top, bottom and full-body garments, (ii) *shoes* in common styles, (iii) *jewelries*, including bracelets, earrings, necklaces and rings, (iv) *accessories*, including bags, belts, hats, glasses, sunglasses and ties.

We consider detailed sub-types if necessary, such as the class *bag* consisted of the backpack, shoulder and tote bags. *Clothes* are divided into top cloth, bottom cloth, and dress. Each sub-type contains two gender groups (woman and man), with the exceptions that *jewelries* and *dress* exclusively contain woman samples, while *tie* contains only man samples.

Each gender group includes 15 model images, where the garments are categorized into three settings: white background, natural background, and try-on setting. Every garment setting include 5 images. Following previous work's categorization of virtual try-on scenarios into *in-shop* and *in-the-wild*, we further divide the model images for *clothes* and *shoes* into 15 shop-style and 15 wild-style samples per gender group, resulting in 30 model images per sub-type.

The benchmark predominantly sources images from public repositories (Pexels[2]), supplemented with brand website materials and social media content under compliant data usage protocols.

**Pairing Strategy.** For each gender group, we establish combinatorial pairs between model and garment images through:

- *Maximum Pair Calculation*: $max\_pairs = 15 \times 15 \times 17 + 30 \times 15 \times 7 = 6,975$ pairs, where 17 and 7 denote model settings counts for regular and style-specific categories respectively.

---

[2] https://www.pexels.com

- *Sampled Pair Selection*: $selected\_pairs = 15 \times 15 \times 24 = 360$ paired samples, constrained by single-use garment policy and balanced sampling (15 models per clothes/shoes type, include 7 shop-style and 8 wild-style).

Overall, our experiments are all evaluated on the selected benchmark contains 360 pairs of images

## A.2 Evaluation Metrics

As discussed before, the objectives of try-on can be divided into three aspects. Since there is no ground-truth result in mask-free setting, we redesign the metrics as follows:

*Object Consistency*: We crop the objects from the try-on and object images via masking, then perform white-background normalization on the extracted objects. We compute the visual similarity using DINO [3] and CLIP [49] visual encoders, with metrics denoted as M-DINO and M-CLIP-I. As these metrics measure cosine similarity in the embedding space, their values range in $[-1, 1]$ where higher values indicate better object preservation. The M-DINO scores generally exhibit lower values than M-CLIP-I, as DINO-extracted features are more sensitive to geometric variations compared to CLIP's semantic-aligned embeddings. Our experiments quantitatively validate this behavior across different object categories. This discrepancy stems from their distinct learning objectives:

- **M-DINO** [3]: Learns dense local features through self-supervised distillation, emphasizing spatial consistency of object parts. Then compute the cosine similarity of two features.
- **M-CLIP-I** [49]: Optimizes global semantic alignment between object images, prioritizing category-level coherence. Then compute the cosine similarity of two features. Then compute the cosine similarity of two object features.

*Person Preservation*: We extract the person regions by cropping try-on and original person images, masking the target object areas with black pixels. We then compute spatial-aligned similarity between these aligned image pairs using two complementary metrics:

- **SSIM** (Structural Similarity Index) [57]: Measures structural, luminance, and contrast similarity between images. The metric ranges in $[-1, 1]$ with values approaching 1 indicating higher structural consistency.
- **LPIPS** (Learned Perceptual Image Patch Similarity) [65]: Computes deep feature differences using pretrained VGG networks, better aligning with human perception than traditional metrics. Its values lie in $[0, 1]$ where lower scores denote better preservation quality.

*Object Localization*: We propose a dual-strategy evaluation framework to assess spatial rationality through complementary approaches:

- **G-Accuracy**: Quantifies detection reliability using GroundingDINO [36] with the following implementation protocol: Invoke *predict_with_classes* API with target object categories as *classes* parameter. Configure detection thresholds: *box_threshold* $= 0.25$ (bounding box confidence) and *text_threshold* $= 0.25$ (text-image alignment). Last, calculate success rate as total test cases correct detections.
- **CLIP-I**: Evaluates semantic alignment through multi-modal similarity measurement: Generate descriptive prompts via Qwen2 [1] MLLM. Compute CLIP [49] embedding similarity between try-on images and generated text. Normalize scores to [0,1] range using min-max scaling.
  The final prompt template is formally defined as follows:

  ```
  """Generate a detailed description of a composite image by combining
  elements from the two provided images:
      1. Image 1: The model's appearance (pose, clothing, facial featur-
      es), background and style
      2. Image 2: Only the <{garment_class}>, without any other infos
      (e.g., background, model)
  Describe the synthesized image with the model wearing the {garment_cl-
  ass}, in 65 words. Only describe the final imagined scene, without the
  detail or information of composite. The main description is from
  ```

```
Image 1. Briefly and shortly describe the {garment_class} in 6 wor-
ds, no details needed. No words like (e.g., from the Image 2). If
{garment_class} is cloth or dress , the model from the Image 1, re-
place with the {garment_class} from Image 2, no words like (replace
the hair/shirt), using "wear" the {garment_class}.

Examples outputs:
    - "A young woman standing in a studio with a white background.
    She is wearing a denim dress with a button-down collar and long
    sleeves. The dress is knee-length and falls above her knees. T-
    he woman is also wearing black ankle boots with a pointed toe
    and a low heel. She has a brown crossbody bag with a strap acr-
    oss her shoulder. The bag appears to be made of leather and has
    a small flap closure. The overall style of the outfit is casual
    and minimalistic."
    - "A close-up portrait of a young woman's face and upper body.
    She is wearing a black strapless top with a thin silver chain n-
    ecklace around her neck. Her hair is styled in loose waves and
    she is wearing large hoop earrings. The woman is looking off to
    the side with a serious expression on her face. The background
    is plain white."
    - "A close-up portrait of a woman's upper body. She is wearing
    a black collared shirt with a button-down collar and long slee-
    ves. Her hair is styled in loose curls and she is wearing large,
    dangling earrings. Her hand is resting on her chest, with a lar-
    ge ring on her ring finger. The background is plain white. The
    woman appears to be looking off to the side with a serious expr-
    ession on her face."
"""
```

# B  Details of Training Dataset

## B.1  Dataset for Stage-1

The model in the first stage is jointly trained on two datasets, *i.e.,* the unpaired in-the-wild images, and the dataset of stage-2 without the object image. We train on the datasets with sampling ratios of 2 : 1. To further investigate the class distribution in the unpaired dataset, we count the highly-frequent words in the object text descriptions. After filtering out the prepositions and verbs, the top-5 words are necklace, hat, glasses, sunglasses and watch. We also observe some classes excluded in our final 12 common classes, *e.g.,* smartphone, cup, scarf, crown and mask. The rich distribution of wearable or holdable objects enhances the generalization of `OmniTry` to uncommon classes.

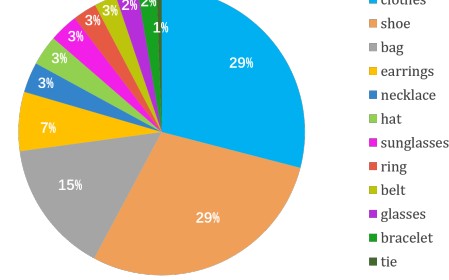

Figure 8: The class distribution of training dataset.

We also report the scale of dataset during the data preparation. The initial dataset contains 152K in-the-wild images, which are filtered to be 111K images with person and wearable objects. After listing, grounding and removing objects, the total amount of images containing at least one object is 94K, and the corresponding number of objects is 189K (roughly 2 objects per image).

## B.2 Dataset for Stage-2

For the training dataset of the second stage, we visualize the amount of samples for each class in Appendix B . It is shown that the most common classes, *i.e.,* clothes and shoes, constitute more half of the total dataset, while most classes lay in the long-tail of distribution with less than $3\%$. Such a distribution is aligned with our basic assumption that it is hard to obtained paired samples for many wearable objects. For class-balanced training, we manfully assign the sampling weights for clothes, shoes and bags as $4, 4, 3$, and set weights as $1$ for remaining classes.

# C  Details of Training and Model Architecture

## C.1  Training Configuration

During training, we resize the image with fixed aspect ratio to be no larger than $1$ million, which means that the model could receive images with varying aspect ratios in one batch. To handle this, we pad the image tokens into the same length of sequence, and modify the attention block to forward only on the valid tokens.

For both training of stage-1 and stage-2, we set the learning rate as $1^{-4}$, gradient accumulation steps as 1, weight decay as $0.01$ and gradient norm clipping as $1.0$. We use the AdamW [38] optimizer with hyper-parameters $\beta_1 = 0.9$ and $\beta_2 = 0.999$. The model is trained with mixed precision of bfloat16. We note that since we fine-tune based on the distilled version of FLUX [32], the guidance scale is fixed as 1 during training, and set as 30 during inference.

## C.2  Details of Re-purposing Inpainting Model

We elaborate the details of adapting the inpainting model, FLUX.1-Fill in this paper, towards mask-free try-on task. During training, the input of model can be split into two sets in sequence dimension:

- The try-on image. Along the channel dimension, it contains the noisy ground-truth try-on image, the input person image and a zero mask in the same shape.

- The object image. Along the channel dimension, it contains the noisy object image, the clean noisy image and a zero mask.

Then during the inference stage, we initialize the above input while replacing the noisy latents with standard Gaussian noise. Through the above formulation, it is shown that the inputs of person and object images are different. The person branch aims to modify the input person image in proper area, while the object branch simply targets to maintain the input, and transfers the object appearance via full attention mechanism.

## C.3  Details of Masked Full-Attention

We discuss the details of applying masked full-attention in the second stage. We set text prompts for both try-on and object images, like "trying on sunglasses". Suppose the length of tokens to be: $L_{I1}$ for try-on image, $L_{T1}$ for try-on text, $L_{I2}$ for object image, and $L_{T2}$ for object text. We concatenate all tokens in the above order. Then the attention mask is:

$$\begin{bmatrix} 1_{L_{I1} \times L_{I1}} & 1_{L_{I1} \times L_{T1}} & 1_{L_{I1} \times L_{I2}} & 0_{L_{I1} \times L_{I1}} \\ 1_{L_{I1} \times L_{I1}} & 1_{L_{I1} \times L_{T1}} & 0_{L_{I1} \times L_{I2}} & 0_{L_{I1} \times L_{I1}} \\ 0_{L_{I1} \times L_{I1}} & 0_{L_{I1} \times L_{T1}} & 1_{L_{I1} \times L_{I2}} & 1_{L_{I1} \times L_{I1}} \\ 0_{L_{I1} \times L_{I1}} & 0_{L_{I1} \times L_{T1}} & 1_{L_{I1} \times L_{I2}} & 1_{L_{I1} \times L_{I1}} \end{bmatrix}, \tag{2}$$

where $1_{m \times n}$ denotes all-one matrix and $0_{m \times n}$ denotes all-zero matrix. More specifically, we apply such a full-attention in both the multi-modality blocks and single blocks of FLUX [32], and figure out the text tokens to achieve the masking. We leverage the attention function with varying length in FlashAttention [12] to implement the block-wise masked attention.

## C.4 LoRA Implementation

We implement the location and identity adapters with LoRA [21]. In detail, we set the rank and $\alpha$ to be 16. We insert the LoRA module into the following layers: the projection into query/key/value, output projection of attention, the linear layers in feedforward block, the layer normalization layer, the input patch projection, and the final output projection.

# D    Details of Compared Methods

In this section, we present the details of compared methods and our implementation of them on try-on task. We also report more results of the variants of each method, among which we only report the best result in main experiment.

## D.1    General Customized Image Generation

**OneDiffusion** [33]: A large-scale diffusion framework supporting bidirectional image synthesis across tasks. We evaluated its performance on mask-free/mask-based try-on through instruction-based cases. We also modify its original instructing prompt to achieve better performance.

**OmniGen** [58]: A vision-language unified framework consolidating multiple tasks, supporting both mask-free/mask-based generation. We also test it with both standard and our optimized prompts.

**VisualCloze** [35] implements visual in-context learning for domain generalization. We conduct experiments with single example and multiple examples in the context.

**Paint-by-Example** [61] enables to re-paint a given subject into image via CLIP-based object representation with mask dependency.

**MimicBrush** [5] achieves imitative inpainting for region-specific edits, requiring the input image with mask, together with the reference image without mask.

**ACE++** [40] extends long-context conditioning for instruction-driven generation that tackles various.

## D.2    Image-based Virtual Try-On

**OOTDiffusion** [60] designs a two-branch U-Net architecture to consume the person and garment images, which requires masked input in the person branch.

**Magic Clothing** [4] introduces a garment extractor to progressively insert garment features into the main backbone of try-on generation. Magic Clothing supports the input of either masked person image, or the targeting pose and person ID image. We adapt the former setting to better preserve the person image.FI

**CatVTON** [10] proposes to transfer the identity of garment by simply concatenating it with the person image, and achieve mask-based try-on with inpainting model.

**FitDiT** [25] introduces diffusion transformer (DiT) model into VTON, and designs a GarmentDiT and a DenoisingDiT to implement this task.

**Any2AnyTryon** [18] is the only open-source mask-free VTON model, eliminates the dependence on masks, poses, or any other such conditions.

## D.3    More Comparison Results

We report more comparison results in Tab. 3, including variants of methods with mask/mask-free setting, varying image size and different prompt design. We report only the best result of all variants in the main experiment.

# E    More Visualization Results

We visualize more try-on results in Fig. 9, where we include all classes in OmniTry-Bench and different sub-types for full visualization.

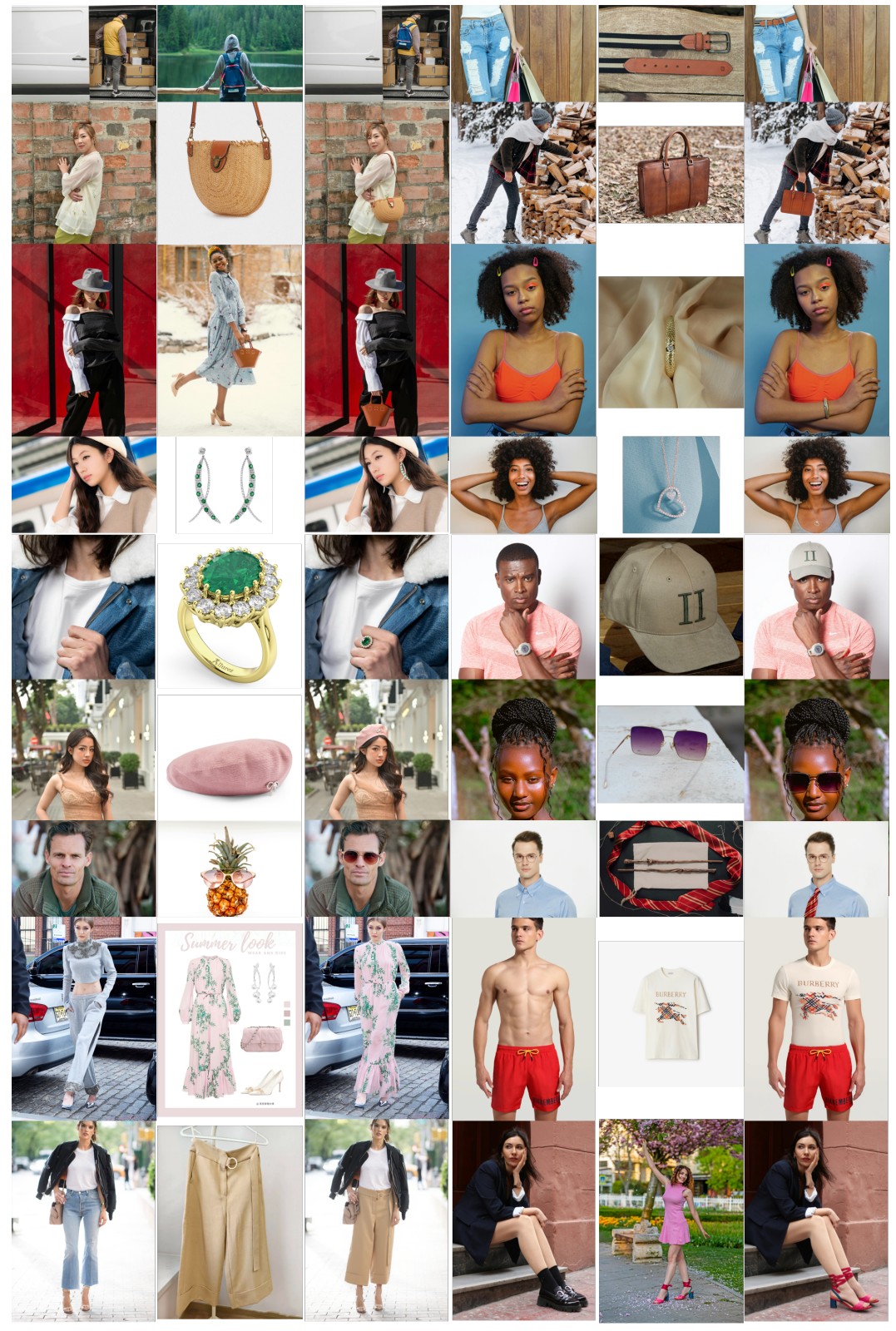

Figure 9: The samples of the model, the object, and the try-on person.

Table 3: More evaluation results of the compared methods with different settings.

| method | mask | Object Consistency | | Person Presevation | | Object Localization | |
|---|---|---|---|---|---|---|---|
| | | M-DINO ↑ | M-CLIP-I ↑ | LPIPS ↓ | SSIM↑ | G-Acc. ↑ | CLIP-T ↑ |
| *on the whole set* | | | | | | | |
| Paint-by-Example ($512^2$) [61] | | 0.4171 | 0.7328 | 0.4577 | 0.7968 | 0.9833 | 0.2831 |
| Paint-by-Example ($1024^2$) [61] | | 0.4565 | 0.7727 | 0.3903 | 0.8033 | 0.9861 | 0.2804 |
| MimicBrush [5] | | 0.4693 | 0.7253 | 0.3033 | 0.8575 | 0.9250 | 0.2781 |
| ACE++ (prompt v1) [40] | | 0.4565 | 0.7474 | 0.4561 | 0.7519 | 0.9667 | 0.2791 |
| ACE++ (prompt v2) [40] | ✓ | 0.4449 | 0.7427 | 0.4554 | 0.7517 | 0.9722 | 0.2793 |
| VisualCloze (1-example) [35] | | 0.4705 | 0.7533 | 0.6685 | 0.5320 | 0.9972 | 0.2283 |
| VisualCloze (2-example) [35] | | 0.4236 | 0.7307 | 0.6767 | 0.4908 | 0.9917 | 0.2260 |
| OmniGen (prompt v2) [58] | | 0.5151 | 0.7761 | 0.6888 | 0.5870 | 0.9917 | 0.2557 |
| OneDiffusion (prompt v1) [33] | | 0.5515 | 0.8137 | 0.6607 | 0.6166 | **1.0** | 0.2290 |
| OneDiffusion (prompt v2) [33] | | 0.5580 | 0.7950 | 0.5795 | 0.6628 | 0.9972 | 0.2401 |
| OneDiffusion (prompt v1) [33] | | 0.4178 | 0.7358 | 0.7606 | 0.4951 | **1.0** | 0.2309 |
| OneDiffusion (prompt v2) [33] | | 0.4731 | 0.7749 | 0.7001 | 0.5831 | 0.9972 | 0.2309 |
| VisualCloze (1-example) [35] | | 0.5292 | 0.7782 | 0.4471 | 0.6190 | 0.9639 | 0.2524 |
| VisualCloze (2-example) [35] | ✗ | 0.4915 | 0.7619 | 0.4730 | 0.5868 | 0.9806 | 0.2540 |
| OmniGen (prompt v1) [58] | | 0.5299 | 0.7689 | 0.7009 | 0.5727 | 0.9778 | 0.2533 |
| OmniGen (prompt v2) [58] | | 0.5435 | 0.7869 | 0.6703 | 0.5965 | 0.9944 | 0.2535 |
| OmniTry (**Ours**) | | **0.6160** | **0.8327** | **0.0542** | **0.9333** | 0.9972 | **0.2831** |
| *on the clothes subset* | | | | | | | |
| Magic Clothing [4] | | 0.5665 | 0.7634 | 0.2761 | 0.8786 | 1.0 | 0.2700 |
| CatVTON [10] | | 0.5744 | 0.7906 | 0.1664 | **0.9283** | 1.0 | 0.2818 |
| CatVTON (w. garment mask) [10] | | 0.5534 | 0.7843 | 0.2084 | 0.8828 | 1.0 | 0.2797 |
| OOTDiffusion [60] | ✓ | 0.5961 | 0.8016 | 0.2178 | 0.8865 | 1.0 | 0.2761 |
| FitDiT ($768 \times 1024$) [25] | | 0.6718 | 0.8324 | 0.1972 | 0.8952 | 1.0 | 0.2822 |
| FitDiT ($1152 \times 1536$) [25] | | 0.6733 | 0.8340 | 0.1618 | 0.9027 | 1.0 | 0.2831 |
| FitDiT ($1536 \times 2048$) [25] | | 0.5961 | 0.8016 | 0.2178 | 0.8865 | 1.0 | 0.2761 |
| Any2AnyTryon [18] | ✗ | 0.6747 | 0.8537 | 0.2089 | 0.8969 | 1.0 | **0.2832** |
| OmniTry (**Ours**) | | **0.6995** | **0.8560** | **0.1021** | 0.9105 | **1.0** | 0.2799 |

Table 4: Human evaluation results of OmniTry and garment-only methods.

| Method | Magic Clothing | CatVTON | OOTDiffusion | FitDiT | Any2AnyTryon | OmniTry (**Ours**) |
|---|---|---|---|---|---|---|
| **Avg. Rank** ↓ | 4.27 | 3.36 | 3.70 | 2.28 | 0.77 | **0.62** |

# F  Human Evaluation of the Generated Try-ons

We conduct a human evaluation to assess the realism and usefulness of the generated try-on results, especially in comparison with garment-only methods. Specifically, we invite five annotators to rank the outputs of different methods based on three aspects: try-on success rate, garment consistency, and overall realism. The average ranking results are summarized in Tab. 4, where a lower value indicates a better ranking. As shown, OmniTry achieves the best overall performance among all compared methods.

# G  Differences between Stage-1 of OmniTry and Editing Methods

The key differences between the stage-1 of OmniTry and the editing methods that support the "Add" operation can be summarized as follows. (1) Task and performance: The general editing methods typically involve a wide range of editing tasks, thus may show restricted performance on specific operation, especially on try-on cases requiring fine-grained combination of the added object and the original image. The added object could be more likely to be an independent item, while OmniTry focuses on natural combination with parts of input person. (2) Method: The stage-1 of OmniTry is designed by re-purposing an inpainting-based model to mask-free editing, leveraging its ability of

Table 5: Compraison between `OmniTry` (stage-1) and editing methods supporting "Add" operation.

| Method | LPIPS | SSIM | G-Acc. | CLIP-T |
|---|---|---|---|---|
| **AnyEdit** | 0.1112 | 0.8455 | 0.8167 | 0.2415 |
| **OmniGen** | 0.3381 | 0.6394 | 0.9889 | **0.2654** |
| `OmniTry` **(stage-1)** | **0.0711** | **0.8959** | **0.9944** | 0.2613 |

Table 6: Additional ablation study on one-stream vs. two-stream adapter.

| Method | trainable params. | M-DINO | M-CLIP | LPIPS | SSIM | G-Acc. | CLIP-T |
|---|---|---|---|---|---|---|---|
| two-stream | 172M (2 LoRA with r=16) | **0.5845** | **0.8159** | **0.0425** | 0.9403 | 0.9806 | **0.2620** |
| one-stream | 172M (1 LoRA with r=32) | 0.5619 | 0.8149 | 0.0439 | **0.9478** | **0.9861** | 0.2604 |

detailed local editing. Specifically, the original image and generated image are concatenated in the channel dimension. However, the general editing methods require larger divergence between the input and output, and are concatenated in the sequential dimension (e.g., UniReal [7] and OmniGen [58]), showing higher computation cost (2x sequence length).

We compare AnyEdit [26] and OmniGen [58] with the stage-one model of OmniTry in Tab. 5, with the metrics of object localization and person preservation. We observe that OmniGen could not guarantee to preserve the original image (similar to its performance in stage-2). For AnyEdit, though it preserve the input image, it could sometimes fail to add any object (worse G-Acc.) or properly combine the object onto the person. We will also include visualization result in revised version.

## H   Additional Ablation Study on One-Stream vs. Two-Stream Adapter

To further ensure the alignment of trainable parameters, we train a new location adapter with double LoRA rank (r=32) from stage-1, and initialize it into the second stage for one-stream training. We note that the additional computation cost is doubled than two-stream adapters with r=16. We initialize both settings from an earlier checkpoints of stage-1 with the same training steps to ensure fair comparsion. The results in Tab. 6 show that though with less computation cost, the two-stream setting still shows better performance to seperately cope with different capabilities of `OmniTry`.

## I   More Discussion on Unexpected Shortcut in Stage-1

In stage-1, we observe that using naively erased training samples leads the model to produce output images that almost perfectly recover the position and shape of the object in the ground-truth image. We hypothesize that this phenomenon is likely caused by information leakage, for the following reasons. (1) The reconstruction shown in Fig. 3 primarily reflects shape and position reconstruction, rather than appearance reconstruction. Since no object image is provided to the first stage, the model generates objects with diverse appearances but consistently reproduces the same shape and position as the ground-truth object. This observation suggests that the model might exploit the boundary of the erased region, enabling it to perfectly reconstruct the object's location and outline. (2) A stronger piece of evidence is observed when we train the model with traceless erasing under the same number of training steps. In this case, the model produces objects with random shapes, positions, and appearances, even when evaluated on the training samples, indicating that the shape recovery in the naive erasing setup indeed stems from boundary leakage.

