# OpenReview forum: "OmniTry: Virtual Try-On Anything without Masks"
_NeurIPS.cc/2025/Conference — NeurIPS 2025 poster_

### Official Review · Reviewer_R2aq · 2025-06-19

**Clarity:** 2
**Significance:** 3
**Originality:** 3
**Rating:** 5
**Confidence:** 4

**Summary:**

This paper presents OmniTry, a virtual try-on framework designed for various wearable objects, featuring a two-stage training process. It is first trained on a large amount of unpaired data, followed by fine-tuning with a small set of paired data, effectively addressing the challenge of obtaining paired images. In addition, OmniTry adopts a mask-free setting, eliminating the need for additional mask-based guidance. Extensive experiments show the superiority of the method in terms of both generation quality and generalization ability.

**Questions:**

It is noted that object images are mostly collected from in-the-wild sources. How does the presence of multiple objects in one image affect the results? For example, in the last case shown in Figure 1, the object image contains a hat, but it is not transferred to the final result.

**Ethical Concerns:**

["NO or VERY MINOR ethics concerns only"]

**Final Justification:**

I have carefully read the authors's rebuttal. Most of my concerns have been addressed by the authors. Therefore, I would like to raise my rating.

**Limitations:**

Yes.

**Paper Formatting Concerns:**

No obvious formatting issues found.

**Quality:**

3

**Strengths And Weaknesses:**

***Strengths:***
1. The paper breaks the conventional limitation of virtual try-on methods to specific target categories and proposes a more general framework.
2. The paper presents diversity and high-quality experimental results.

***Weaknesses:***
1. The reliance on multiple preprocessing steps such as text description generation, segmentation, and object erasing may lead to significant error accumulation, which could negatively affect the final performance of the model.
2. It is confusing that the final output of Traceless Erasing is a processed try-on image, while the information leakage problem is initially attributed to the erased image. The training configuration following the traceless erasing step is not clearly described.
3. The description of the mask in Figure 4 as "optional" appears ambiguous, implying that OmniTry can also generate results based on these masks. Additionally, corresponding text descriptions are not provided in the samples.
4. Example results showcasing varying transfer effects on the same object, as described in line 126, are missing from the paper.
5. Typo found at Line 112.

---

> ### Author Rebuttal · Authors · 2025-07-30
>
> Thanks for your careful reading and valuable feedback. We provide the following responses for your concerns:
>
> **[W1] The reliance on multiple preprocessing steps such as text description generation, segmentation, and object erasing may lead to significant error accumulation, which could negatively affect the final performance of the model.**
>
> **[A1]** Firstly, the preprocessing steps are all based on general models verified and commonly adopted by the community, *i.e.*, QwenVL for object description, Grounding-SAM for object segmentation and FLUX-FIll for object erasing. We also fine-tune the FLUX-Fill to further improve its ability of erasing. Therefore, the final processed data show great available rate for usage. Furthermore, the second stage is based on strictly-defined 12 classes with accurate text description, which ensures the final model to be stable on the defined-tasks.
>
> Secondly, based on the chain of processing, later step can filter out failure cases of previous step instead of accumulating errors. For instance, when the MLLM generate descriptions of objects that does not exist in image, the later grounding model could return an empty list of bounding boxes by setting a proper threshold.
>
> Thirdly, since the processed images serve as the input of model while the natural images serve as output supervision, the wrongly-processed images can be viewed as input augmentation, and improve the robustness of model to deal with imperfect input images.
>
> ---
>
> **[W2] It is confusing that the final output of Traceless Erasing is a processed try-on image, while the information leakage problem is initially attributed to the erased image. The training configuration following the traceless erasing step is not clearly described.**
>
> **[A2]** We apologize for misleading the understanding of the final output from traceless erasing. In Figure 3 (b), the final output is a pair of images, *i.e.*, **the final try-on image and the final erased image**. The whole process of traceless erasing and the following training configuration are explained as follows:
>
> - Since we use image-to-image translation to eliminate the leaked erasing boundary, the unedited area is also slightly changed compared with the original try-on image. Directly training with such un-matched pairs will lead to worse person preservation. Therefore, we carefully blend the final erased image with the original try-on image regarding the object mask, and output the final try-on image with precise alignment in the unedited area with erased image.
> - During training, the final erased image serves as the input of model, while the final output image serves as the supervision of output. Throughout this way, the erasing traces in original erased images will not affect the training process with information leakage.
>
> We will make the presentation more clear in the revised version.
>
> ---
>
> **[W3] The description of the mask in Figure 4 as "optional" appears ambiguous, implying that OmniTry can also generate results based on these masks. Additionally, corresponding text descriptions are not provided in the samples.**
>
> **[A3]** Thanks for your suggestion.
> * For the mask, we will modify the figure to distinguish the methods with/without masks in the revised version. Currently, please refer to Table 1 to distinguish the compared methods.
> * For the corresponding text descriptions, we directly use the pre-defined prompt like “trying on bag/belt/hat/…” and “replacing the top cloth” in stage-2, as suggested in line 220-222. We will also add the descriptions into the figure in revised version.
>
> ---
>
> **[W4] Example results showcasing varying transfer effects on the same object, as described in line 126, are missing from the paper.**
>
> **[A4]** Thanks for your suggestion. Firstly, we distinguish varying transfer effects in the text description of stage-1, which helps the model to learn more precise localization ability. In the main experiments on OmniTry-Bench, we concentrate on the most common try-on effects of each class. To further exploring the remaining transfer effects, we conduct studies on uncommon classes as shown in Sec. 4.4, where we showcase results of different interactions, like holding something. Secondly, to better showcase varying transfer effects learned in stage-1, we will add an visualization in revised version, showing different try-on types with the same type of object, *e.g.,* wearing sunglasses and holding sunglasses.
>
> ---
>
> **[W5] Typo at line 112.**
>
> **[A5]** Thanks for pointing it out. We will modify it and carefully polish the writing of paper in revised version.
>
> ---
>
> **[W6] How does the presence of multiple objects in one image affect the results? For example, in the last case shown in Figure 1, the object image contains a hat, but it is not transferred to the final result.**
>
> **[A6]** For object images containing multiple objects, the referring object is controlled by the textual prompt. In Figure 1, we use “trying on the bag” and the model only put on the bag in object image. We apologize for missing the description, and will add the text description in revised version.

---

> > ### Author Response · Authors · 2025-08-06
> >
> > Dear Reviewer,
> >
> > We would like to gently remind you that we have submitted the rebuttal for addressing your concerns on our submission. We sincerely appreciate your effort on reviewing our work and sharing valuable feedback. If there are any remaining questions on any points, please feel free to reach out. We would be glad to engage in the discussion and endeavor to address your concerns.
> >
> > Thank you for your time and consideration.

---

> ### Comment · Reviewer_R2aq · 2025-08-09
>
> Thank the authors for their rebuttal. I have carefully read the rebuttal and most of my concerns have been addressed. So I would like to remain my initial rating.

---

> > ### Author Response · Authors · 2025-08-09
> >
> > Dear Reviewer,
> >
> > We truly appreciate your feedback and the time you spent reviewing our work. It is encouraging to know that our clarification has addressed most of your concerns. We will further polish our paper following your suggestions.
> >
> > Thanks for your time and consideration.

---

### Official Review · Reviewer_UUZt · 2025-06-27

**Clarity:** 3
**Significance:** 2
**Originality:** 2
**Rating:** 3
**Confidence:** 5

**Summary:**

This paper presents OmniTry, a unified mask-free framework for virtual try-on (VTON) across a wide range of wearable objects beyond garments, such as accessories and jewelry. The authors propose a two-stage training pipeline: the first stage learns mask-free localization using unpaired images and object descriptions, while the second stage introduces a small amount of paired data to enforce object identity consistency. A comprehensive benchmark covering 12 object categories is constructed to evaluate performance under both in-shop and in-the-wild conditions.

The paper demonstrates a substantial engineering effort and addresses a practically valuable problem in the field of image-based try-on systems. The breadth of supported object types, the construction of a new evaluation benchmark, and the overall system integration reflect a significant amount of implementation and empirical work.

However, I have concerns regarding the novelty (Section 3.2: Unpaired Data Pre-process, Model Architecture) and the effectiveness (Section 3.3: Stage-2: ID Consistency Preservation) of the proposed approach. Incorporating more comparative experiments with existing methods, as well as ablation studies on the proposed components, would make the paper more complete.

**Questions:**

1.How does the unpaired data construction process proposed in this work differ from the "Add" operation unpaired data construction commonly used in existing image editing methods, such as AnyEdit, UniReal, and OmniGen? Could the authors include comparative experiments to better highlight the uniqueness of their strategy?

2.Given that the inpainting model architecture closely resembles existing subject-driven image editing frameworks such as In-Context-LoRA, what are the specific innovations in the proposed model design? Would it be possible to add comparisons with structurally similar baselines?

3.Is the Two-Stream Adapter design necessary, considering that the second-stage model alone achieves comparable performance with sufficient paired samples? Could a unified single-adapter design be equally effective, and if so, has the parameter efficiency been fairly compared in Table 2?

4.Why does disabling full attention lead to better LPIPS scores in Table 2? The explanation provided in lines 288–289 is insufficiently supported. Can the authors include attention visualizations and additional qualitative analysis to clarify the impact of full attention?

**Ethical Concerns:**

["NO or VERY MINOR ethics concerns only"]

**Final Justification:**

Thanks for addressing my concerns. The task specific for virtual try-on anything without masks is interesting and I appreciate for the efforts of boosting the perfomance. However, I am not fully convinced by the rebuttal towards the technical contributions against previous works. Therefore, I decide to keep my original rating.

**Limitations:**

1.This work still does not address the critical issue of how clothing size affects the try-on results, which remains a key challenge in virtual try-on.

2.The proposed method may experience performance degradation in scenarios where the image contains multiple persons or when occlusions are present.

**Paper Formatting Concerns:**

None.

**Quality:**

4

**Strengths And Weaknesses:**

Strengths:

Quality:

1.This paper proposes a unified mask-free virtual try-on framework for multiple object categories, which demonstrates good usability and try-on quality.

2.The engineering implementation is thorough and well-documented, providing useful insights for the virtual try-on task.

3.This paper introduces a benchmark that evaluates virtual try-on models from multiple perspectives, contributing to the advancement of the try-on research community.

Weaknesses:

Originality:

1.The unpaired data construction method described in Section 3.2 (Unpaired Data Pre-process) is a commonly used approach in recent literature, corresponding to the "Add" operation in image editing tasks. Similar strategies can be found in AnyEdit (CVPR 2025), UniReal (CVPR 2025), and OmniGen (CVPR 2025). Including comparisons with these methods could better highlight the distinctiveness of the proposed approach.

2.The Inpainting Model presented in line 133 under Model Architecture is a widely adopted backbone for subject-driven image editing. Similar architectures are used in In-Context-LoRA, as well as in references [1] and [2]. I suggest adding comparisons with methods based on similar structures to more clearly demonstrate the differences and advantages of your design.

Clarity & Significance:

1.Figure 5 does not indicate the specific training cost of the first stage. Moreover, the figure shows that comparable performance can be achieved using only 200 samples without any first-stage training. Considering the additional computational cost of the first stage, it seems plausible that training with only the second stage could also be acceptable.

2.Building upon the point above, the necessity of the Two-Stream Adapter design is unclear. It may be feasible to jointly train on unpaired and paired samples using only a single LoRA adapter. Additionally, for the one-stream adapter ablation in Table 2, the parameter count of the one-stream adapter should be reported. It is important to ensure that the one-stream and two-stream setups are comparable in both parameter size and training effort.

3.Table 2 shows that the best LPIPS score is achieved without using full attention. What explains this phenomenon? The explanations in lines 288–289 are insufficiently supported. I recommend adding more attention visualizations and qualitative results to clarify the role and effectiveness of full attention.

Reference:

[1] Large-Scale Text-to-Image Model with Inpainting is a Zero-Shot Subject-Driven Image Generator. (CVPR2025)

[2] Flux Already Knows – Activating Subject-Driven Image Generation without Training.

[3] In-Context-LoRA: IN-CONTEXT LORA FOR DIFFUSION TRANSFORMERS https://github.com/ali-vilab/In-Context-LoRA

[4] AnyEdit: Mastering Unified High-Quality Image Editing for Any Idea.

[5] UniReal: Universal Image Generation and Editing via Learning Real-world Dynamics.

[6] OmniGen: Unified Image Generation.

---

> ### Author Rebuttal · Authors · 2025-07-30
>
> Thanks for your careful reading and valuable feedback. We provide the following responses for your concerns:
>
> ---
>
> ### Originality:
>
> **[W1] Comparison between the stage-1 of OmniTry and general editing models supporting “Add” operation.**
>
> **[A1]** Thanks for the suggestion. For the referenced works (AnyEdit, UniReal and OmniGen), since UniReal is close-sourced, we compare AnyEdit and OmniGen with the stage-one model of OmniTry, with the metrics of object localization and person preservation. We observe that OmniGen could not guarantee to preserve the original image (similar to its performance in stage-2). For AnyEdit, though it preserve the input image, it could sometimes fail to add any object (worse G-Acc.) or properly combine the object onto the person. We will also include visualization result in revised version.
>
> | method | LPIPS | SSIM | G-Acc. | CLIP-T |
> | --- | :---: | :---: | :---: | :---: |
> | AnyEdit | 0.11117 | 0.8455 | 0.8167 | 0.2415 |
> | OmniGen | 0.3381 | 0.6394 | 0.9889 | **0.2654** |
> | OmniTry (stage-1) | **0.0711** | **0.8959** | **0.9944** | 0.2613 |
>
> We further conclude the difference between OmniTry stage-1 and "Add" editing models as follows:
>
> - **Task and performance**: The general editing methods typically involve a wide range of editing tasks, thus may show restricted performance on specific operation, especially on try-on cases requiring fine-grained combination of the added object and the original image. The added object could be more likely to be an independent item, while OmniTry focuses on natural combination with parts of input person.
> - **Method**: The stage-1 of OmniTry is designed by re-purposing an inpainting-based model to mask-free editing, leveraging its ability of detailed local editing. Specifically, the original image and generated image are concatenated in the channel dimension. However, the general editing methods require larger divergence between the input and output, and are concatenated in the sequential dimension (e.g., UniReal and OmniGen), showing higher computation cost (2x sequence length).
>
> ---
>
> **[W2] Comparison between the inpainting-based subject-driven editing methods.**
>
> **[A2]** The key difference is that we conduct different type of task. The referenced methods (*e.g.*, In-Context LoRA and FLUX already knows) is to **conduct subject-driven generation instead of editing**. They generally concatenate the generating image with the input image in spatial dimension, and set an inpainting mask with the generating region to be all 1. The model is trained to generate a new image referring its connected input image. Thus, they achieve subject-driven generation where the output contains the subject with different poses or in varying scenes. In contrast, the task of OmniTry is local editing where the input person image is preserved in unedited region. Since the models are designed for different task setting, they could not be directly compared with OmniTry.
>
> ---
>
> ### Clarity & Significance:
>
> **[W1] On the ablation of two-staged training**
>
> **[Q1-1]** The lack of training cost of the first stage in Figure 5.
>
> **[A1-1]** The training cost of stage-1 is presented in the implementation details of Sec 4.1, showing 50K steps of training on unpaired dataset. We will re-clarify it in the discussion of ablation study in the revised version.
>
> **[Q1-2]** Considering the comparable performance using 200 samples without any first-stage, training with only the second stage could also be acceptable.
>
> **[A1-2]** We disagree. Firstly, we note that the 200 samples in Figure 5 indicates 200 samples **per class.** Though it can be easily obtained for classes like clothes, it could be challenging to collect paired data for most long-tailed classes, such as ties and gloves. Experimental results suggest that the re-purposing inpainting model for mask-free localization takes much more efforts than the second stage. Therefore, it is worthy of leveraging the high computation cost on large-scale unpaired data in stage-1, and contributing a model that can be easily adapted by few samples and less training cost. We would make the checkpoints of both stages to be public, where the community could fine-tune from the first checkpoint on their own classes of objects with a few samples.
>
> Secondly, the current OmniTry is evaluated on pre-defined 12 common classes of wearable objects. However, the construction of the dataset in stage-1 covers a much wider range of classes, e.g., watches and earphones. This enables us to efficiently fine-tune to uncommon classes as shown in Figure 6. If the model was only trained with the second stage of strictly defined 12 classes, it would be challenging to be extended to other classes.
>
> ---
>
> **[W2] On the necessity of two-stream adapter design.**
>
> **[Q2-1]** Is the Two-Stream Adapter design necessary, considering that the second-stage model alone achieves comparable performance with sufficient paired samples? Could a unified single-adapter design be equally effective?
>
> **[A2-1]** Firstly, there might be some misunderstanding on the experimental setup and the relationship between two-staged training and two-stream adapter in OmniTry. The results with/without stage one in Figure 5 are all based-on two-stream adapter, while the result of “one-stream adapter” in Table 2 indicates using two-staged training and directly fine-tuning the location-adapter in the second stage. Therefore, it already show the result of using single adapter with sufficient data in both stages, demonstrating the effectiveness of two-stream design. We apologize for the unclear description and will make the settings more clear in the revised version.
>
> Secondly, from the perspective of method design, the two adapters serve for different purposes in OmniTry. The location adapter learns to repurpose the inpainting model for mask-free object insertion, while the identity adapter learns to transfer the appearance of objects. Though using a single adapter could also achieve comparable performance, the generalized localization ability studied from the first stage would be forgotten when fine-tuned with specific classes in stage-2, which restricts further extending to more classes.
>
> Thirdly, since the two-stream adapters deal with different tokens of input, the total computation cost is exactly the same as using a single adapter. Considering the parameters of LoRA module is relatively small, it is acceptable to use two separate adapters with more flexible performance.
>
> **[Q2-2]** The parameter count of the one-stream adapter should be reported ensuring comparable setups.
>
> **[A2-2]** Thanks for the suggestion. The two-stream and one-stream settings in Table 2 are all based on the same location adapter from stage-1 for fair comparison. The two-stream setting adds a new identity adapter for training, and one-stream continually fine-tunes the location adapter. Therefore, it would be impossible to align the trainable parameters when ensuring the same initialization. Since the two-stream adapters deal with separated tokens from person and object images, the computation cost is strictly aligned for comparison.
>
> To further ensure the alignment of trainable parameters, we train a new location adapter with double LoRA rank (r=32) from stage-1, and initialize it into the second stage for one-stream training. We note that the additional computation cost is doubled than two-stream adapters with r=16. Due to time limit, we could not train the new location adapter with r=32 in aligned training steps of stage-1 with Table 2. Thus, we initialize both settings from an earlier checkpoints of stage-1 with the same training steps to ensure fair comparsion. The final results will be added in the revised version.
>
> | method | trainable params. | M-DINO | M-CLIP-I | LPIPS | SSIM | G-Acc. | CLIP-T |
> | --- | --- | :---: | :---: | :---: | :---: | :---: | :---: |
> | two-stream | 172M (2 LoRA with r=16) | **0.5845** | **0.8159** | **0.0425** | 0.9403 | 0.9806 | **0.2620** |
> | one-stream | 172M (1 LoRA with r=32) | 0.5619 | 0.8149 | 0.0439 | **0.9478** | **0.9861** | 0.2604 |
>
> The results show that though with less computation cost, the two-stream setting still shows better performance to seperately cope with different capabilities of OmniTry.
>
> ---
>
> **[W3] Why does disabling full attention lead to better LPIPS scores in Table 2?**
>
> **[A3]** We firstly apologize for the misleading, and note that the “full attention” in Table 2 indicates **using full attention** instead of disabling. We will make it clear in revised version. The reasons why it achieves better LPIPS are as follows:
>
> - Better LPIPS indicates the non-object area is better preserved. The original OmniTry blocks the attention from person image to object image with masking. When using full attention without the above operation, the information of person image is emphasized, thus showing higher person preservation metrics.
> - However, the person preservation metrics can not be independently viewed. When the input person image is completely copied without adding any objects, the best LPIPS with 0 will be achieved.  Adding too small or incomplete objects could also lead to better LPIPS. Thus, the metrics should be evaluated together with the consistency and localization metrics.
> - Since no image can be attached in rebuttal, we will add the attention visualization showing the information flow from person to object images in the revised version.
>
> ---
>
> ### Limitations:
>
> **[W1] Limitation on adapting clothing size and multiple/occluded persons.**
>
> **[A1]** Thanks for pointing out the additional limitation of our work. The clothing size and multiple/occluded person input are all important topics for virtual try-on, which can be further addressed in the future works of this paper. We will add them in the limitation discussion of our revised version.

---

> > ### Author Response · Authors · 2025-08-06
> >
> > Dear Reviewer,
> >
> > We would like to gently remind you that we have submitted the rebuttal for addressing your concerns on our submission. We sincerely appreciate your effort on reviewing our work and sharing valuable feedback. If there are any remaining questions on any points, please feel free to reach out. We would be glad to engage in the discussion and endeavor to address your concerns.
> >
> > Thank you for your time and consideration.

---

### Official Review · Reviewer_JoQf · 2025-06-30

**Clarity:** 3
**Significance:** 2
**Originality:** 3
**Rating:** 4
**Confidence:** 3

**Summary:**

This paper proposes a unified framework that enables virtual try-on of multiple wearable objects. Specifically, the proposed method first uses large-scale unpaired images to perform prior learning. This step is essentially a detect-erase-restore process, where multi-modal large language models are introduced to provide descriptions of all wearable items as generative guidance. Then, the model is further fine-tuned using paired data conditioned on the corresponding wearable object images.

**Questions:**

See “Weaknesses” in Review.

**Ethical Concerns:**

["NO or VERY MINOR ethics concerns only"]

**Limitations:**

Yes.

**Quality:**

3

**Strengths And Weaknesses:**

**Strengths**
1.  This paper proposes a novel framework that supports the virtual try-on for 12 common classes of wearable objects, which is an interesting and sound idea.
2.  This paper is well-structured, and the experiments are clearly described.

**Weaknesses**
1. Although the design of the framework is technically sound, the use of multiple preprocessing steps and pre-trained models makes the overall pipeline complex and computationally expensive. The trade-off between the effectiveness and efficiency needs to be further considered.
2. The paper attributes the perfect reconstruction in erased samples to information leakage. Is there sufficient evidence to support this claim? Could it also be caused by other factors, such as overfitting to the training data?
3. The example in Figure 3(b) does not clearly demonstrate the effectiveness of Traceless Erasing. Could the authors provide more convincing results, such as the necklace case in Figure 3(a)?

---

> ### Author Rebuttal · Authors · 2025-07-30
>
> Thanks for your careful reading and valuable feedback. We provide the following responses for your concerns:
>
> **[W1] The use of multiple preprocessing steps and pre-trained models makes the overall pipeline complex and computationally expensive.**
>
> **[A1]** Firstly, the multiple pre-processing steps with pre-trained models are all used the in the data preparation stage, thus the final try-on model is efficient with end-to-end generation manner. Secondly, the pre-processing pipeline is designed as an automatic process with stable models verified by the community (QwenVL, Grounding-SAM and FLUX-Fill). Similar data preprocessing pipeline has been adapted in works like UNO [1] and UniReal [2].
>
> To further verify the computation cost of data pre-processing, we compute the averaged runtime of each steps as follows. It is noted that the result is the sum of processing all objects in an image with a single H800 gpu process, which is acceptable for processing the whole dataset.
>
> | pre-processing step | MLLM (QwenVL-7B) | GroundingDINO+SAM | Traceless Erasing (FLUX-Fill) | Overall |
> | --- | :---: | :---: | :---: | :---: |
> | avg. runtime (s/image) | 0.87 | 0.83 | 15.82 | 17.52 |
>
> ---
>
> **[W2] The paper attributes the perfect reconstruction in erased samples to information leakage. Is there sufficient evidence to support this claim? Could it also be caused by other factors, such as overfitting to the training data?**
>
> **[A2]** We appreciate the valuable suggestion, and make it more clear to verify the information leakage as follows:
> * The reconstruction in Figure 3(a) refers to **shape/position reconstruction** instead of appearance reconstruction. Since no object image is fed into the first stage, we observe that the model randomly generate varying appearance of object while always maintaining the same shape as the ground-truth object in the same position of person. Such an observation suggests that the boundary of the erased area is leaked for model to perfectly reconstruct the shape and position.
> * A more obvious evidence is that when we train the model on data with traceless erasing in the same training steps, the model shows random shape/position/appearance of objects when evaluated on training samples. This demonstrates that the reconstruction is caused by information leakage instead of training set overfitting. We will add the visualization and detailed discussion in the revised version.
>
> ---
>
> **[W3] The example in Figure 3(b) does not clearly demonstrate the effectiveness of Traceless Erasing. Could the authors provide more convincing results, such as the necklace case in Figure 3(a)?**
>
> **[A3]** Thanks for the suggestion. Since the erasing traces is noted as invisible, it would be hard to observe obvious difference between the processed images and input images, as shown in the area of red boxes in Figure 3(b). To demonstrate the effectiveness of traceless erasing, we conduct ablation study and show results in Table 2, where traceless erasing greatly improve the try-on performance. Furthermore, as discussed in [A2], we will also include the visualization examples on processed data similar to Figure 3(a) in the revised version.
>
>
> ---
>
> References:
>
> [1] Less-to-More Generalization: Unlocking More Controllability by In-Context Generation
>
> [2] UniReal: Universal Image Generation and Editing via Learning Real-world Dynamics

---

> > ### Author Response · Authors · 2025-08-06
> >
> > Dear Reviewer,
> >
> > We would like to gently remind you that we have submitted the rebuttal for addressing your concerns on our submission. We sincerely appreciate your effort on reviewing our work and sharing valuable feedback. If there are any remaining questions on any points, please feel free to reach out. We would be glad to engage in the discussion and endeavor to address your concerns.
> >
> > Thank you for your time and consideration.

---

### Official Review · Reviewer_GR1r · 2025-06-30

**Clarity:** 3
**Significance:** 3
**Originality:** 2
**Rating:** 4
**Confidence:** 4

**Summary:**

OmniTry presents a unified, mask-free virtual try-on framework that extends beyond garments to any wearable object via a two-stage training pipeline: Mask-free localization on large-scale unpaired portrait images, using MLLM-guided object description, object removal, and an inpainting-based diffusion transformer repurposed to learn where and how to paint objects. ID-consistency fine-tuning on paired in-shop images, introducing an identity adapter and masked full-attention to preserve object appearance 4326

**Questions:**

1. Can OmniTry adapt to novel wearable categories that were not part of the training set, and how many paired examples are necessary for effective adaptation?
2. How does the model perform when the descriptions generated by the MLLM are incorrect or noisy? Is there a degradation in localization accuracy or identity preservation?

**Ethical Concerns:**

["NO or VERY MINOR ethics concerns only"]

**Limitations:**

See Weakness and Questions

**Quality:**

3

**Strengths And Weaknesses:**

Strengths
1. To my knowledge, it is the first framework to support any wearable object, not just clothing.
2. The two-stage training pipeline leverages unpaired real-world images for localization, improving its generalization performance.
3. The results compared with current works seems to be reasonable.

Weakness
1. This work involves many modules or strategies, like two-stage training, but the core contributions and novelties are not clearly presented. It seems it is a combinations of variant modules and lack original novelty.
2. Any2AnyTryon seems to address a similar task. The relationship and differences with this work should be presented.
3. It depends on external modules. The reliance on MLLM for object descriptions and erasing models means that errors in these stages can affect the final output, leading to potential inaccuracies.
4. The paper does not include a human evaluation to assess how realistic or useful the generated try-ons are, especially when compared to garment-only methods.

---

> ### Author Rebuttal · Authors · 2025-07-29
>
> Thanks for your careful reading and valuable feedback. We provide the following responses for your concerns:
>
> **[W1] The core contributions and novelties are not clearly presented.**
>
> **[A1]** Besides the new task exploration and evaluation, we summarize the core technical novelties of OmniTry as follows:
>
> - It presents a two-staged framework solution towards multi-condition generation scenarios where training samples with complete conditions are hard to be obtained. OmniTry leverages large-scale unpaired data that can be more easily obtained to achieve mask-free local editing. Such a framework benefits that: (i) less paired samples are required for training to achieve comparable performance (*e.g.*, 10 samples per class), (ii) the model can be easily extended to uncommon classes with various try-on effects.
> - It proposes to re-purpose the inpainting model for mask-free editing tasks to leverage its intrinsic property of input preservation, in contrast to general editing models based text-to-image models. Besides, since the generating noisy image is stacked with the input in channel dimension, OmniTry achieves more efficient computation compared with general mask-free editing methods (with 1.5x sequence length of tokens).
> - The invisible information leakage of traditional data pre-processing is observed and verified by experiments. Then we design the traceless erasing to effectively improve the try-on success rate, which can also be used for improving existing mask-free VTON methods.
> - We design a two-stream architecture to separately cope with the localization and consistency transferring problem, which properly serves the two-stage training framework.
>
> ---
>
> **[W2] The relationship and differences with Any2AnyTryon.**
>
> **[A2]** The differences are as follows:
>
> - **Task**: The main difference is that we focus on different try-on tasks. OmniTry focuses on unified try-on with a wide range of  wearable object classes, while Any2AnyTryon focuses on clothing try-on under varying settings, *e.g.*, normal try-on, model-free try-on and try-on in layers.
> - **Challenge**: OmniTry address the problem where paired training samples are hard to be obtained for many uncommon classes of objects, while Any2AnyTryon is based on sufficient training samples and aims to improve the try-on performance.
> - **Method**: OmniTry is based on inpainting model where the input person image and generating try-on image are stacked in channel dimension. Any2Any2Tryon is based on text-to-image model and concatenate condition images with generating noises in spatial dimension. Thus, the inference cost of Any2Any2Tryon is 1.5 times of OmniTry with longer sequence length.
>
> ---
>
> **[W3] The external modules, MLLM for object descriptions and erasing models means that errors in these stages can affect the final output.**
>
> **[A3]** Firstly, the external modules for data pre-processing are all generally examined by the community, *i.e.*, QwenVL as MLLM for describing objects, Grounding-SAM as segmentor, and FLUX-Fill as erasing model. We also fine-tune the FLUX-Fill to enhance its erasing ability. Therefore, the final processed data show great available rate for training. Furthermore, the second stage is based on strictly-defined 12 classes with accurate text description instead of generated text, which ensures the final model to be stable on the defined-tasks.
>
> Secondly, the later module in the preprocessing can help to filter out failure cases of previous module. For instance, when the MLLM generate descriptions of objects that does not exist in image, the later grounding model could return an empty list of bounding boxes by setting a proper threshold.
>
> Thirdly, since the processed image serves as the input of model while the natural image serves as output supervision, the wrongly-processed images can be viewed as input augmentation for training and improve the model’s robustness to deal with imperfect input images.
>
> ---
>
> **[W4] The paper does not include a human evaluation to assess how realistic or useful the generated try-ons are, especially when compared to garment-only methods.**
>
> **[A4]** Thanks for your suggestion. We have conducted a human evaluation comparing OmniTry with the garment-only methods. Due to time limit, we invite 5 annotators to evaluate the outputs of these methods, and annotate the rankings of them based on a comprehensive evaluation of try-on success rate, garment consistency and realistic output. The results are as follows, where OmniTry achieves the best result compared with existing methods.
>
> | method | Magic Clothing | CatVTON | OOTDiffusion | FitDiT | Any2AnyTryon | OmniTry (Ours) |
> | --- | :---: | :---: | :---: | :---: | :---: | :---: |
> | Avg. Rank | 4.27 | 3.36 | 3.70 | 2.28 | 0.77 | **0.62** |
>
> We will include more detailed human evaluation on other classes in the revised version.
>
> ---
>
> **[W5] Can OmniTry adapt to novel wearable categories that were not part of the training set, and how many paired examples are necessary for effective adaptation?**
>
> **[A5]** For supporting a novel category, there are two requirements: (i) The category exists or is similar to examples in the unpaired dataset of stage-1. Since the unpaired dataset only requires a single image of trying-on the object, we have curated a large-scale set of samples to cover most classes of wearable objects with different interactions, covering the uncommon wearable items like hairband and gloves, and holdable items like phones and books. (ii) Paired data fine-tuning. Results Sec. 4.4 show that around 20 paired samples can efficiently adapt the model to a novel category.
>
> ---
>
> **[W6] How does the model perform when the descriptions generated by the MLLM are incorrect or noisy? Is there a degradation in localization accuracy or identity preservation?**
>
> **[A6]** Firstly, the object description generated by MLLM is used for object grounding and segmentation in the follow-up process. Thus, the wrong descriptions can be filtered by set a grounding threshold, representing that the described object does not exist in the image. Secondly, the descriptions in the second stage is not generated by MLLM, but strictly divided into different classes with a unified textual indication, like "trying on the sunglasses". Therefore, the description will not influence the final performance of unified try-on.

---

> > ### Author Response · Authors · 2025-08-06
> >
> > Dear Reviewer,
> >
> > We would like to gently remind you that we have submitted the rebuttal for addressing your concerns on our submission. We sincerely appreciate your effort on reviewing our work and sharing valuable feedback. If there are any remaining questions on any points, please feel free to reach out. We would be glad to engage in the discussion and endeavor to address your concerns.
> >
> > Thank you for your time and consideration.

---

### Note · Authors · 2025-08-12

We sincerely appreciate all the reviewers and the AC for their careful reading and valuable feedback on our submission. We are encouraged that OmniTry is considered with novel framework, interesting and sound idea, well-structured presentation, and diverse experiments with new benchmark.

During the discussion period, we have addressed the concerns of reviewers as follows:

- **Data Pre-processing:** Explanation on the stability and robustness of data pre-processing, and the computation cost of the overall pipeline.
- **Technical Contribution:** Re-clarify the technical contribution of OmniTry and its difference with related works including try-on, general editing and subject-driven generation methods, with more quantitative comparison.
- **Human Evaluation:** Add a human evaluation ranking results compared with existing methods.
- **Effectiveness of Designed Modules:** Add results and explanation on modules including traceless erasing, two-stage training and two-stream adapter design.
- **Other Details:** Polish the presentation including descriptions of Figure 1&4 and typo at line 112.

Besides the aforementioned points, we will also add **visualization results** in the revised version, including training monitoring after traceless erasing and attention weights of using full attention.

While we understand that no further interaction is possible at this stage, we hope that our responses have sufficiently addressed all concerns. We respectfully request the reviewers and AC to consider whether our clarifications and proposed revisions have resolved the issues raised during the review. We remain fully committed to incorporating all suggestions to improve our paper for the final version. Thank you for your time and consideration throughout this process.

---

### Decision · Program_Chairs · 2025-09-17

**Decision:**

Accept (poster)

**Comment:**

The submission received 4 thorough reviews and was deeply discussed.
The rebuttal and additional answers help clarify some concerns.
Ultimately, the reviewers reached a consensus and agreed to recommend an Accept. Congratulations!

We encourage the authors to include all the comments the Reviewers suggested.
Please particularly improve clarification of the technical contributions against previous works, as suggested by the reviews.